# Recovery of homogeneous photocatalysts by covalent organic framework membranes

Hao Yang [1,3], Jinhui Xu[2,3], Hui Cao[2], Jie Wu[2] ✉ & Dan Zhao[1] ✉

Transition metal-based homogeneous photocatalysts offer a wealth of opportunities for organic synthesis. The most versatile ruthenium(II) and iridium(III) polypyridyl complexes, however, are among the rarest metal complexes. Moreover, immobilizing these precious catalysts for recycling is challenging as their opacity may obstruct light transmission. Recovery of homogeneous catalysts by conventional polymeric membranes is promising but limited, as the modulation of their pore structure and tolerance of polar organic solvents are challenging. Here, we report the effective recovery of homogeneous photocatalysts using covalent organic framework (COF) membranes. An array of COF membranes with tunable pore sizes and superior organic solvent resistance were prepared. Ruthenium and iridium photoredox catalysts were recycled for 10 cycles in various types of photochemical reactions, constantly achieving high catalytical performance, high recovery rates, and high permeance. We successfully recovered the photocatalysts at gram-scale. Furthermore, we demonstrated a cascade isolation of an iridium photocatalyst and purification of a small organic molecule product with COF membranes possessing different pore sizes. Our results indicate an intriguing potential to shift the paradigm of the pharmaceutical and fine chemical synthesis campaign.

Visible-light-induced photocatalysis has witnessed drastic developments in organic chemical synthesis over the past two decades[1–3]. Noble metal-based chromophores such as ruthenium(II) and iridium(III) polypyridyl complexes represent the most versatile and effective photoredox catalysts. However, both ruthenium (Ru) and iridium (Ir) are rare elements, comprising only $10^{-3}$ ppm of the earth's crust[1,4,5]. The scarcity and high price of these noble metals have made it challenging to use them widely, especially in large-scale industrial settings. The immobilization of these photocatalysts on solid resins is usually impractical because of obstructed light transmission or low catalyst loading capacity of the solid resins. There have been extensive efforts to replace noble metal-based photocatalysts with more economical and sustainable photocatalysts, such as those comprising earth-abundant metals[6] and organic dyes[7]. Unfortunately, the low efficiency associated with short excited state lifetime[8] and poor photo/chemical stability of these catalysts remains an obstacle to their widespread adoption. Therefore, there is a need to develop a robust, energy-saving, and environmentally-friendly process for homogeneous photocatalyst recycling.

Membrane-based nanofiltration, a pressure-driven membrane separation process, could be feasible for homogeneous photocatalyst recycling[9]. Compared to conventional separation techniques, nanofiltration is operationally simpler and has lower energy demand and smaller spatial requirements[10–12]. in addition, nanofiltration prevents damage to heat-sensitive molecules (vs. evaporation and distillation) and acid-sensitive molecules (vs. silica-gel-based chromatography). Existing nanofiltration technologies mostly rely on size exclusion-based separation using either ceramic or polymeric membranes.

[1]Department of Chemical and Biomolecular Engineering, National University of Singapore, 4 Engineering Drive 4, 117585 Singapore, Singapore. [2]Department of Chemistry, National University of Singapore, 3 Science Drive 3, 117543 Singapore, Singapore. [3]These authors contributed equally: Hao Yang, Jinhui Xu. ✉e-mail: chmjie@nus.edu.sg; chezhao@nus.edu.sg

Although ceramic membranes feature good chemical stability, they are expensive, and there remain challenges in terms of scalability and structural modification[13]. In contrast, dense polymeric membranes can be readily processed and scaled up, forming the majority of membranes with industrial applications. Very recently, Noël's group reported an in-line photocatalyst recovery protocol by continuous flow and nanofiltration using polymeric membranes[14]. Nonetheless, dense polymeric membranes are limited by their low-flux nature and poor chemical resistance to organic solvents.

In recent years, various advanced porous polymer materials, such as covalent organic frameworks (COFs), have been developed based on reticular chemistry[15–17]. Unlike amorphous polymers, COFs are crystalline due to their highly ordered structures[18]. Therefore, COFs with controllable pore size and highly regulated crosslinked frameworks are promising candidates for constructing advanced separation membranes[19]. The customized pore sizes, consisting of atoms arranged in myriad structures with specific dimensions, are especially appealing for the nanofiltration of small molecules[20,21]. Notably, the membranes can be optimized according to the three-dimensional molecular size of the target molecules. However, successful applications of COFs in homogeneous catalyst recovery have yet to be attained due to inadequate developments in organic solvent-resistant COF membranes. At present, the use of COF membranes is mainly limited to water treatment[22–24].

Herein, we describe the fabrication of solvent-resistant COF membranes with customized pore sizes for the effective recovery of homogeneous photocatalysts. To demonstrate the broad applicability of these COF membranes, the photocatalyst recovery performance for six homogeneous photocatalysis reactions is investigated. The resulting COF membranes possess tunable pore sizes and remarkably high organic solvent resistance, exhibiting high catalyst recovery rates and long-term stability towards a wide range of organic solvents. The permeance of these COF membranes is two orders of magnitude higher than the conventional polymeric membranes for catalyst recovery. A gram-scale recovery of the photocatalyst is achieved, demonstrating the practicality of COF-membrane-based nanofiltration. Furthermore, an appealing but challenging cascade separation of reagents, photocatalysts, and products is achieved using COF membranes with different pore sizes. This study may inspire the design of customizable COF membranes and promote advances in the on-demand recovery of homogenous photocatalysts.

## Results

### Preparation of organic solvent-resistant COF membranes

The fabrication of COF membranes usually involves growing a thin COF selective layer onto a polymeric substrate. However, commercial polymeric substrates exhibit low stability in polar organic solvents. Current methods for enhancing the stability of polymeric substrates include treatment with conditioning agents, wet or dry annealing, and crosslinking[25,26]. However, most of these methods pose difficulties in scaling up, together with limited stability enhancement and poor reproducibility. In this study, to improve the chemical resistance of polymeric substrates, we first prepared carbonized polyacrylonitrile (PAN) substrates by carbonizing commercial PAN ultrafiltration membranes under an inert atmosphere (Supplementary Fig. 1). Calcium nitrate was used as a pore-forming agent during the carbonization process to produce large pores and enhance porosity. The PAN substrates were then pyrolyzed at an optimized temperature of 210 °C under air to obtain carbonized PAN substrates with crosslinked structures (Fig. 1a). Notably, the carbonized PAN substrates could be easily scaled up and folded without being ruptured (Supplementary Fig. 2). The asymmetric finger-like pores remained after the carbonization process, as shown in its cross-sectional field-emission scanning electron microscopy (FESEM) image (Fig. 1b). The surface FESEM image shows that pores with sizes of 100–500 nm were generated (Fig. 1c).

The carbonized PAN substrates were expected to provide superior fluxes for solvent permeation. Consequently, the solvent resistance and permeance of the carbonized PAN substrates prepared at different temperatures were tested (Supplementary Fig. 3). The volume and weight swelling degrees of the optimized carbonized PAN substrates were then measured after 15 days of soaking in six typical organic solvents (Fig. 1d). The carbonized PAN substrates displayed superior solvent resistance with very low volume swelling (from 0.2 to 3.2%) and weight swelling (from 0.1 to 3.3%). In stark contrast, the uncarbonized PAN substrates showed much higher swelling degrees (>10%) in ethanol, $n$-hexane, and acetone. In addition, they were dissolvable in $N,N$-dimethylformamide (DMF), $N$-methyl-2-pyrrolidone (NMP), and dimethyl sulfoxide (DMSO). The dramatically enhanced solvent resistance of carbonized PAN substrates is mainly due to their structural transition from linear polymeric chains into highly crosslinked network structures with low conformational flexibility (Supplementary Fig. 4).

In a long-term test, the carbonized PAN substrates demonstrated favorable stability in aggressive organic solvents for 60 days, including DMF, NMP, and DMSO (Fig. 1e). The mechanical properties of uncarbonized and carbonized PAN substrates were characterized by the tensile test (stretching rate of 2 mm min⁻¹). The uncarbonized PAN exhibited a tensile strength of 12.6 MPa and Young's modulus of 634.5 MPa (Supplementary Fig. 5). In contrast, the tensile strength of the carbonized PAN decreased by 21%, while Young's modulus increased by 133%. These results demonstrate the high mechanical properties of the carbonized PAN substrates to meet the requirements for practical applications.

Water and organic solvents (both polar and nonpolar) were used to evaluate the solvent permeance of the carbonized PAN substrates (Supplementary Fig. 6). The permeance ($J$) of different solvents as a function of the reciprocal of the viscosity ($\eta$) was plotted, demonstrating a relationship of $J = K/\eta$, in which K is a proportionality constant. This relationship indicates that the carbonized PAN substrates feature hydrophobic and permanent nanopores, and the molecular transport within the carbonized PAN substrates is mainly related to the viscosity of the solvents[27]. The molecular weight cut-off (MWCO) of the carbonized PAN substrate was estimated to be around 1400 kDa (Supplementary Fig. 7).

After the successful preparation of solvent-resistant carbonized PAN substrates, two-dimensional (2D) imine-linked COF membranes were synthesized in situ on the carbonized PAN substrates through interfacial polymerization (Fig. 1f). The aldehyde monomer of 1,3,5-triformylphloroglucinol (Tp) and four different amine monomers (hydrazine hydrate (HZ); 1,3,5-tris(4-aminophenyl)benzene (TAPB); $p$-phenylenediamine (PDA); and 3,3-dihydroxybenzidine (DHBD)) were utilized for the synthesis of COF membranes, which were denoted as Tp-HZ, Tp-TAPB, Tp-PDA, and Tp-DHBD, respectively (Fig. 1g). The pore sizes of COF Tp-HZ, Tp-TAPB, Tp-PDA, and Tp-DHBD are 0.8, 1.2, 1.8, and 2.4 nm, respectively, obtained by the molecular simulation and reported pore size distribution results[28,29]. The COF membranes showed apparent X-ray diffraction (XRD) peaks at lower $2\theta$ values (Tp-DHBD: 3.6°, Tp-PDA: 4.8°, Tp-TAPB: 5.7°, and Tp-HZ: 7.1°), resulting from the corresponding (100) planes (Supplementary Fig. 8). The characteristic XRD patterns confirm their highly crystalline nature.

Surface FESEM images of the COF membranes (Fig. 1h and Supplementary Fig. 9) display dense and void-free surface morphology. Figure 1h shows the surface morphology of a COF Tp-TAPB membrane inserted with its optical image. The cross-sectional FESEM images (Fig. 1i and Supplementary Fig. 10) demonstrate that the COF layer with a thickness of ~100 nm adheres to the carbonized polymeric substrate tightly, benefitting from the in-situ interfacial polymerization. Cross-sectional transmission electron microscopy (TEM) image also shows a similar thickness of the COF layer (Fig. 1j), and the COF lattice was observed with a $d$-spacing of 0.38 nm, corresponding to the (001) plane (Fig. 1k). The chemical structures of these COF membranes were

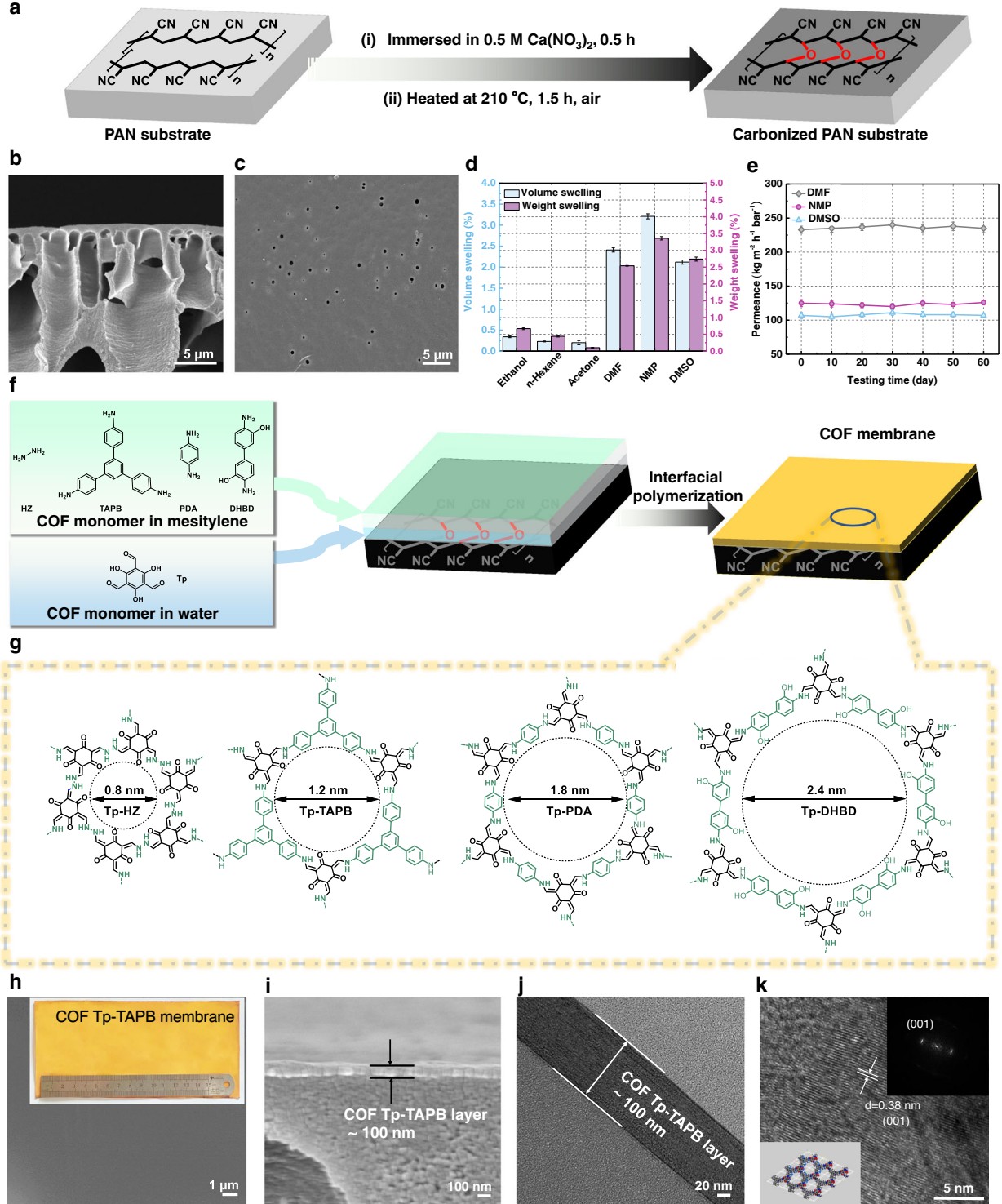

**Fig. 1 | Preparation and characterization of organic solvent-resistant COF membranes. a** Preparation procedure of carbonized PAN substrates. **b**, **c** Cross-sectional and surface FESEM images of carbonized PAN substrates. **d** Volume and weight swelling properties of carbonized PAN substrates in different organic solvents. **e** Long-term organic solvent permeation test of carbonized PAN substrates. **f**, **g** Preparation procedures and chemical structures of COF membranes. **h** Surface FESEM image (inserted with an optical image, and the ruler has a scale up to 15 cm) and **i** Cross-sectional FESEM image of COF Tp-TAPB membrane. **j** Cross-sectional TEM image of COF Tp-TAPB membrane. **k** High-resolution TEM image of COF Tp-TAPB membrane (inserted with a crystal structure image of Tp-TAPB showing the (001) plane (lower left) and a fast Fourier transformation (FFT) image (upper right)). Error bars represent standard deviations for 3 measurements.

characterized by $^{13}C$ solid-state nuclear magnetic resonance (NMR) (Supplementary Fig. 11).

We evaluated the dye rejection performance of these COF membranes to investigate their MWCO using the aqueous solutions of four organic dyes (acid fuchsin (585 Da), Congo red (696 Da), methyl blue (799 Da), and Evans blue (960 Da)) with a concentration of 100 ppm each (Supplementary Fig. 12). The results demonstrate that the Tp-HZ membrane could easily retain the dye molecules and display a high water permeance of 150 kg m$^{-2}$ h$^{-1}$ bar$^{-1}$. Compared with the Tp-HZ membrane, the COF membranes with larger pore sizes have relatively

lower rejection and higher permeance. The pure solvent permeance of the COF Tp-TAPB membrane, which is on the opposite order of the solvent viscosity, is illustrated in Supplementary Fig. 13. Notably, the permeance of the COF membrane towards polar aprotic solvent (e.g., DMF) could be as high as 50 L m$^{-2}$ h$^{-1}$ bar$^{-1}$. The long-term separation performance tests of these COF membranes in DMF indicate the high stability of these COF membranes for separations involving aggressive organic solvents (Supplementary Fig. 14). The performance of our COF membranes is superior to that of the commercial nanofiltration membranes as well as the state-of-the-art thin-film composites (TFC) and graphene oxide membranes (Supplementary Fig. 15)[30,31], thus indicating the potential of these COF membranes for organic solvent nanofiltration.

### Customization of COF membranes for the recovery of photocatalysts

During nanofiltration-based catalyst recycling, molecular weight (MW) enlargement of catalyst/ligand is normally required to generate the MW gap between the catalyst and other solutes for efficient molecular separation[32]. However, the noble metal photocatalysts in this study possess a distinct MW difference ( > 600 Da) from most small organic molecules. Therefore, the customizable COF membranes with tunable pore sizes possess great opportunities for the direct separation of these photocatalysts. To investigate the performance of these as-synthesized COF membranes for recycling metal-based photocatalysts, we selected some well-known Ir- and Ru- photoredox catalysts, including [Ir{dF(CF$_3$)ppy}$_2$dtbbpy]PF$_6$ ([Ir-1]PF$_6$), [Ir(ppy)$_2$dtbbpy]PF$_6$ ([Ir-2]PF$_6$), [Ir{dF(CF$_3$)ppy}$_2$bpy]PF$_6$ ([Ir-3]PF$_6$), and [(Ru(bpy)$_3$](PF$_6$)$_2$ ([Ru](PF$_6$)$_2$), as well as one of the widely utilized hydrogen atom transfer (HAT) photocatalyst, NaDT[1,3]. We targeted to cover the most common reaction types in photocatalysis, including photoredox[33], photo HAT[34], energy transfer (EnT)[35,36], metallaphotoredox[37], and dual catalysis for enantioselective reactions[38]. A chiral Brønsted acid (R)-TRIP was involved in this dual catalytic enantioselective photo transformation, and its separation can indicate whether the current strategy is universal for homogeneous catalyst recycling that goes beyond metal-based photocatalysts.

To test if the COF membranes could withstand different aggressive solvents, we used a wide range of organic solvents in these reactions, including N,N-dimethylacetamide (DMA), hexafluoroisopropanol (HFIP), dichloromethane (DCM), methanol (MeOH), DMSO, acetonitrile (MeCN), tetrahydrofuran (THF), and 1,4-dioxane. Molecular sizes of these catalysts based on the corresponding crystal structures are illustrated in Fig. 2a. Photocatalyst rejection and solvent permeance were initially tested using the as-prepared four kinds of COF membranes (Fig. 2b). Permeance increased along with the pore size of COF membranes. In contrast, rejection showed a decreasing trend with increasing pore sizes of COF membranes. Thus, we selected the optimal COF membranes with high rejection and permeance for each photocatalyst (Supplementary Table 1).

### Recycling of homogenous photocatalysts by COF membranes

We subsequently evaluated the photocatalyst recycling performance of six different homogeneous photocatalysis reactions in conventional batch reactors, including a metallaphotoredox promoted aryl amination (Fig. 3a, Supplementary Figs. 16 to 26 and Supplementary Table 2), an intermolecular [4 + 2] cycloaddition (Fig. 3b, Supplementary Figs. 27 to 36 and Supplementary Table 3), an intramolecular [2 + 2] cycloaddition through EnT (Fig. 3c, Supplementary Figs. 37 to 46 and Supplementary Table 4), methylation of heteroarenes (Fig. 3d, Supplementary Figs. 47 to 56 and Supplementary Table 5), an aromatic C-H thiolation by single-electron transfer (SET) (Fig. 3e, Supplementary Figs. 57 to 66 and Supplementary Table 6), and a dual catalytic enantioselective Minisci-type addition (Fig. 3f, Supplementary Figs. 67 to 77 and Supplementary Table 7). After each reaction, the solution mixture was directly filtered at room temperature through the optimal COF

membrane under argon (2 or 4 bar). The photocatalyst was further washed with the corresponding reaction solvents and then directly subjected to the next reaction cycle. Each reaction-separation process was conducted for 10 cycles, and the reaction yield, enantioselectivity of the product, catalyst recovery rate, and permeance were recorded. These COF membranes enabled constantly high photocatalyst recovery rates ( > 95% in most cases) and high permeance (40-120 kg m$^{-2}$ h$^{-1}$ bar$^{-1}$) in 10 continuous recycling processes. In particular, the dual catalytic Minisci-type reaction afforded high recovery rates for both Ir photocatalyst and organocatalyst (R)-TRIP, constantly obtaining high yields and enantiomeric excess (ee) values of the products. The steady permeance in 10 cycles indicates the good anti-fouling properties of COF membranes. Besides, the COF membranes can be easily regenerated with lifetimes higher than one month, demonstrating high potential for industrial application. Notably, the permeance of these COF membranes is two orders of magnitude higher than the conventional polymeric membranes for catalyst recovery[39].

Degradation of photocatalysts would result in low recovery rates and poor performance in the consequent catalyst-reusing processes. In this study, the recovery rates and reaction yields in 10 cycles were all steady, indicating a high catalytic activity of the recovered photocatalysts. After the final cycle, the purity of the recovered photocatalysts was confirmed by NMR analysis (Supplementary Figs. 21–23, 32–34, 42–44, 52–54, 62–64, 73–75, and 85–87). No detectable photodegradation of catalysts was observed as the NMR ($^1$H, $^{31}$P, $^{19}$F) spectra of the original photocatalysts and the recovered photocatalysts after 10 cycles are identical. In addition, the wavelengths of the UV-Vis absorbance of the recovered photocatalysts after 10 cycles were the same as the original ones (Supplementary Figs. 18, 29, 39, 49, 59, and 69). Therefore, there is no obvious degradation of these noble metal catalysts. It is worth noting that the loss of UV-Vis absorbance with increasing cycle number is due to the slightly decreased concentration of the recovered catalyst solution.

### Gram-scale recovery of photocatalysts by COF membranes

We attempted a gram-scale recovery of the photocatalyst to further demonstrate the practicality of COF-membrane-based nanofiltration. Direct access to γ-fluoroleucine in an operationally simple continuous-flow reactor was achieved through photocatalytic C-H fluorination using NaDT as a photo HAT catalyst[40]. The reaction was performed on a 90 mmol-scale using 2.21 g NaDT (1 mol%). After 16 h of nanofiltration, 2.1 g NaDT was recovered with a COF Tp-DHBD membrane at a 95% recovery rate (Fig. 4a and Supplementary Figs 78 to 81), and then 21.3 g (90% yield) (S)-γ-fluoroleucine was isolated. These results demonstrated the feasibility of COF membranes in handling high-concentration feeds for nanofiltration.

### Stepwise separation of photocatalysts and products by COF membranes

Owing to the modularity of COF membranes, we further attempted to achieve a more appealing but challenging cascade separation of reagents, photocatalysts, and products using COF membranes with different pore sizes. A three-component olefin difunctionalization was achieved using [Ir-3]PF$_6$ and nickel dual catalysis[41]. We first separated [Ir-3]PF$_6$ photocatalyst by a dead-end filtration of the crude mixture through a COF Tp-TAPB membrane with a 96% recovery rate (Supplementary Figs. 82–87). Subsequently, the filtrate was subjected to a COF Tp-HZ membrane to purify the desired product from the remaining starting materials and nickel catalysts with 90% isolated yield (Fig. 4b). High-performance liquid chromatography (HPLC) analysis indicated that the purity of the product was increased from 77% from the crude reaction to 86% after membrane filtration, which also illustrated a very efficient separation of the product from small-molecule starting materials (Supplementary Figs. 88–95). The cascade catalyst recovery and product purification through membrane

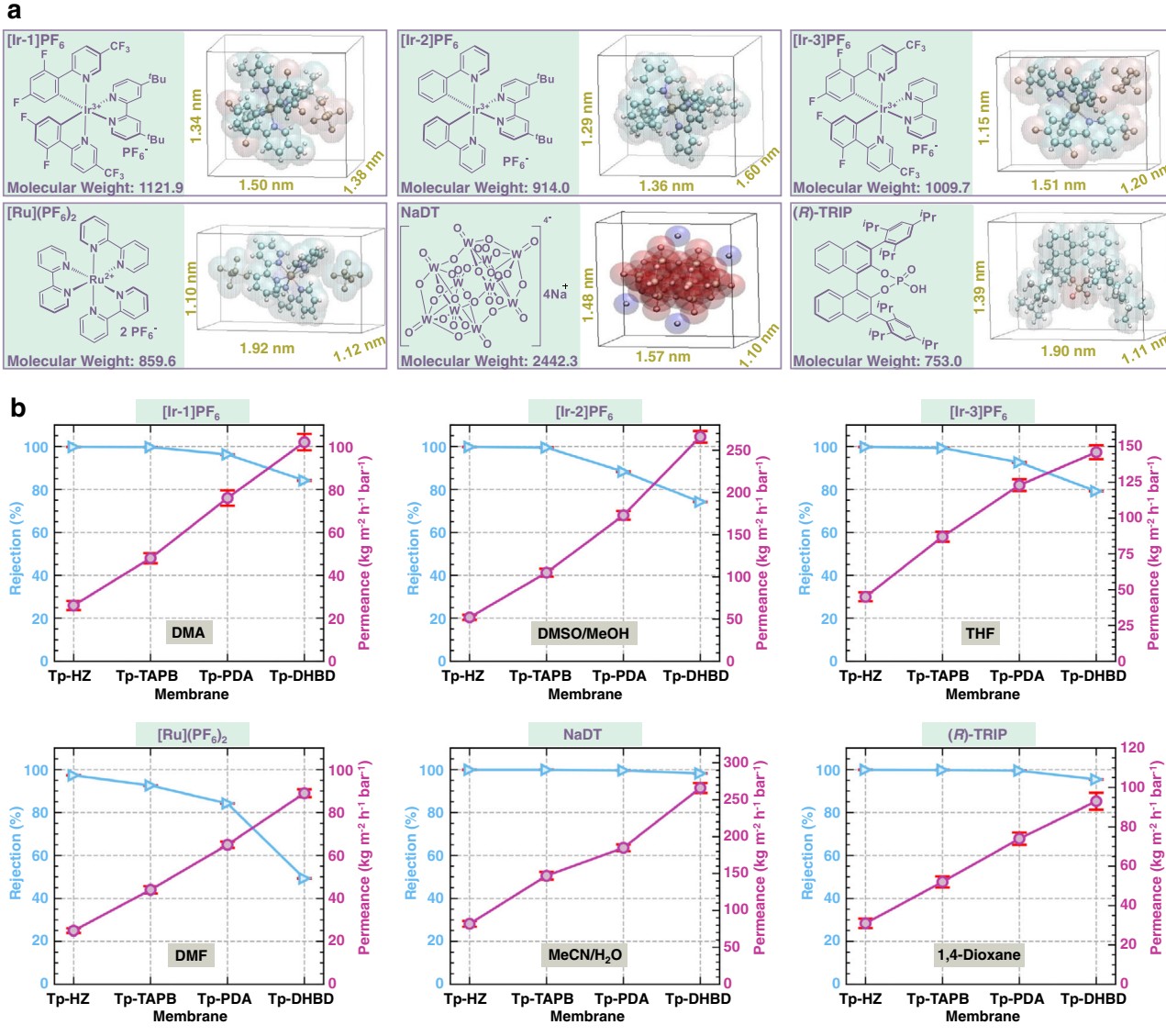

**Fig. 2 | Separation performance of COF membranes. a** Three-dimensional sizes of the crystal structures of [Ir-1]PF$_6$, [Ir-2]PF$_6$, [Ir-3]PF$_6$, [Ru](PF$_6$)$_2$, NaDT, and (*R*)-TRIP catalysts. **b** Separation performance of the customizable COF membranes for the recovery of specific photocatalysts. Error bars represent standard deviations for 3 measurements.

separation is manpower/energy-saving and solvent-economic, representing a superior mode for the downstream operation of homogeneous catalysis.

## Discussion

In summary, this study represents an example of photocatalyst recovery by COF membranes, illustrating the significant advantages of COF membranes in terms of high stability, facile adjustability, and promising catalyst recovery performance. We successfully prepared a series of solvent-resistant COF membranes with customized pore sizes on carbonized polymeric substrates. The effective recovery and reuse of the most widely utilized metal-based photocatalysts from various types of photocatalysis reactions were achieved by dead-end nanofiltrations based on these COF membranes. The permeance of these COF membranes is higher than that of conventional polymeric membranes by up to two orders of magnitude, making them suitable for scalable separation. The customizable pore sizes allow the COF membranes to be optimized according to the three-dimensional size of the targeted catalyst or compound. These COF membranes also exhibit longer lifetime than conventional polymeric membranes, which are usually intolerant of organic solvents and tend to swell. The catalyst recovery from a dual-catalytic system or large-scale flow synthesis was feasible. Moreover, these COF membranes with different pore structures hold great potential for cascade separation/purification of catalysts and products in homogeneous photocatalysis. COF membranes have a high potential for continuous nanofiltration and catalyst recycling. However, the integration of COF membranes into continuous catalyst recycling is still immature compared with conventional polymeric membranes. More in-depth studies on using COF membranes for continuous catalyst recycling will be conducted in the future. The highly customizable structures and high chemical resistance of COFs provide infinite opportunities to develop next-generation synthetic membranes for on-demand recycling of homogeneous photocatalysts and purification of final products, laying a foundation for broader application and adoption of photocatalysis in industrial settings.

## Methods

### Preparation of carbonized polymeric substrates

The carbonized PAN substrates were prepared by carbonization of commercial PAN substrates in a tube furnace. The PAN substrates were

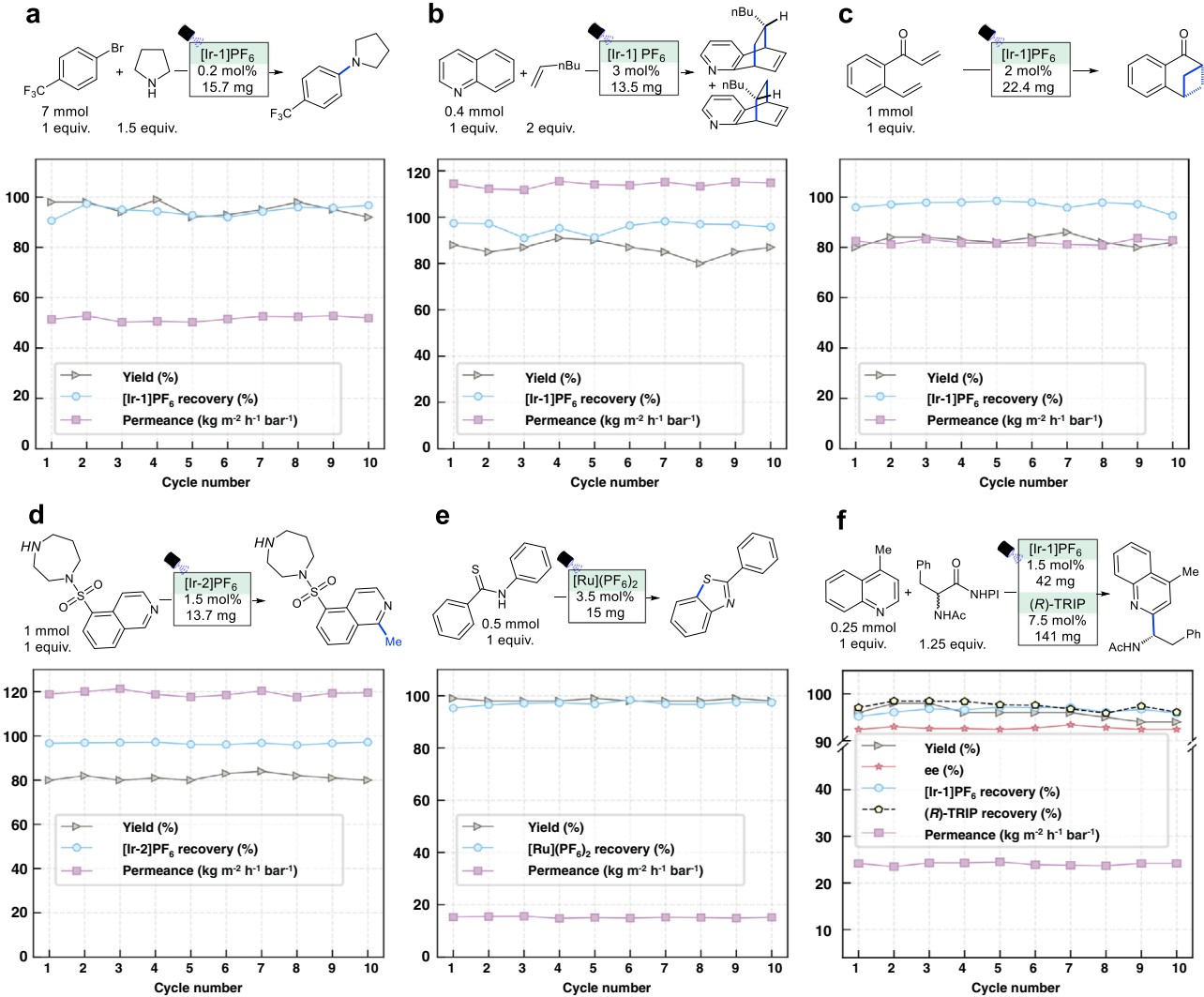

**Fig. 3 | Catalyst recovery performance of different homogeneous photo-catalysis reactions using COF membranes. a** Aryl amination via metallaphotor-edox (solvent: DMA). **b** Intermolecular cycloaddition via EnT (solvent: HFIP), **c** Intramolecular cycloaddition via EnT (solvent: DCM). **d** Alkylation of heteroarenes via SET (solvent: MeOH/DMSO). **e** Aromatic C-H thiolation via SET (solvent: MeCN). **f** Enantioselective Minisci-type addition via SET (solvent: 1,4-dioxane).

first immersed in a 0.5 mol L$^{-1}$ calcium nitrate aqueous solution for 24 h. After drying at room temperature, the PAN substrates were transferred into a tube furnace. The whole carbonization was conducted under an air atmosphere. The temperature inside the tube furnace was first increased from 30 to 210 °C with a ramp rate of 2 °C min$^{-1}$. Then the PAN was carbonized at 210 °C for 90 min. After naturally cooling to room temperature, the carbonized PAN substrates were taken out from the tube furnace and washed with water and ethanol, respectively. The carbonized porous PAN membranes were stored in deionized water for further usage.

## Preparation of COF membranes
The 2D imine-linked COF membranes were in-situ synthesized on the carbonized polymeric substrates through interfacial polymerization. A 20 mL mixture comprising amine monomer (HZ, TAPB, PDA, or DHBD, 1.2 mM), *p*-toluene sulfonic acid (1.0 mM), and water was poured onto the surface of the carbonized substrates. The liquid on the substrates was removed after 1 min, then a 20 mL mixture of Tp (0.9 mM) and mesitylene was poured onto the substrates. The system was kept static for 5 min, and then the liquid on the substrate surface was removed. The as-prepared membranes were then immersed into an acetic acid aqueous solution (2.5 mM) at 60 °C for 36 h. The membranes were taken out from the acetic acid aqueous solution and then

separately washed with ethanol, acetone, and THF to obtain the final COF membranes (denoted as Tp-HZ, Tp-TAPB, Tp-PDA, and Tp-DHBD, respectively).

## Evaluation of solvent permeance and MWCO of carbonized substrates and COF membranes
Solvent permeation performance of the prepared carbonized PAN substrates was evaluated in a dead-end filtration system. Water and common organic solvents were loaded into the system with the pre-pared carbonized PAN substrates to test their solvent permeance. Before the test, the upstream side of the membrane was first kept at 2 bar for at least 2 h to reach a steady state. Then, the permeate was collected and weighed three times at fixed intervals, and the average value of the permeance (*J*, kg m$^{-2}$ h$^{-1}$ bar$^{-1}$) was obtained. MWCO of membranes was measured via rejection experiments using poly-ethylene oxide (PEO) with different molecular weights (Mw = 100k, 300k, 1000k, 2000k, and 5000k Da) at a concentration of 50 ppm as the feed solutions. The concentrations of permeate and feed solutions were determined by a total organic carbon analyzer (TOC ASI-5000A, Shimazu, Japan).

The membrane separation performance was evaluated by calcu-lating rejection and organic solvent flux. Water and common organic solvents were poured into the setup with the prepared COF membranes

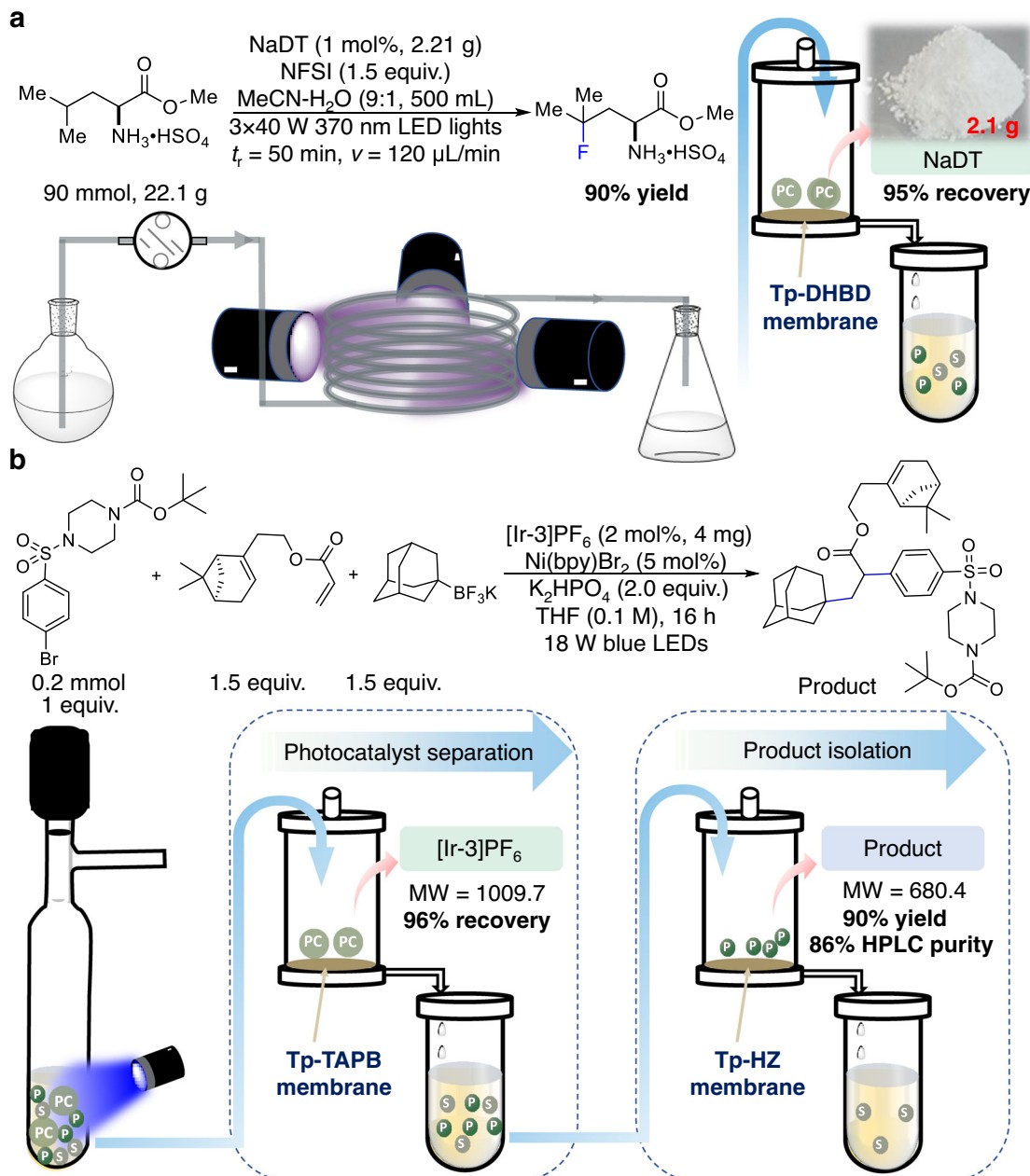

**Fig. 4 | Separation performance of COF membranes for a gram-scale recovery of photocatalysts and a stepwise separation of photocatalysts and products.** **a** Schematic illustration of photocatalyst recovery at 2 g-scale using a COF membrane. **b** Schematic illustration of a stepwise separation of photocatalysts and products using two COF membranes. NFSI, *N*-fluorobenzenesulfonimide; P, product; S, starting material; PC, photocatalyst.

to test their flux. Aqueous solutions containing dyes (100 ppm) were poured into the setup with the prepared COF membranes to test the rejection. The pressure of the feed side was maintained at 2 bar. Before collecting samples, 0.5 h was given to the system for stabilization. The dye rejections were determined by UV-Vis spectroscopy. The permeate samples were collected at least three times to obtain the average values and standard deviations of the final results.

## Customization of COF membranes for recovery of photocatalysts

Screening of the optimal COF membranes for recovery of different photocatalysts was conducted in a dead-end organic solvent nanofiltration system. The performance of the four kinds of COF membranes for rejecting each kind of photocatalyst in organic solvent (same as the corresponding reaction) was evaluated by calculating rejection

and organic solvent permeance. The concentration of the feed was 1000 ppm, and the pressure of the feed side was maintained at 2.0 bar. Before collecting samples, 0.5 h was given to the system for stabilization. The rejections were determined by UV-Vis spectroscopy. The permeate samples were collected at least three times to obtain the average values and standard deviations of the final results.

## Photocatalysis reactions and photocatalyst recovery by COF membranes

**General procedure for aryl amination via SET.** A solution of DMA (70 mL) with photocatalyst [Ir-1]PF$_6$ (15.7 mg, 0.014 mmol, 0.2 mol%), NiBr$_2$•glyme (12.6 mg, 0.35 mmol, 5 mol%), and DABCO (1.41 g, 12.6 mmol, 1.8 equiv.) in a 100 mL round bottom flask (RBF) was bubbled with argon for 15 min. Then 4-bromobenzotrifluoride (980 μL, 7 mmol, 1.0 equiv.) and pyrrolidine (877 μL, 10.5 mmol, 1.5 equiv.) were

added to the above solution. The reaction was irradiated by 18 W blue LEDs under an argon atmosphere and maintained at ambient temperature by cooling with a fan. After 12 h, 1 mL reaction mixture was taken out to examine the yield using dibromomethane as an internal standard. The remaining solution was filtered through the COF Tp-TAPB membrane under an argon atmosphere (2 bar). After the nanofiltration, the COF membrane, together with the retentate, was washed with DMA (5×10 mL) under sonication for 30 min, and the solution of the recovered photocatalyst was obtained. The recovery rate of the photocatalyst was measured by UV-Vis absorption. A total volume of 70 mL of DMA solution containing the recovered photocatalyst was placed in a 100 mL RBF, bubbled with argon for 15 min, and used for the next cycle. Ten cycles of reactions and separations were conducted. After the final cycle, the recovered catalyst was dried and subjected to NMR analysis.

**General procedure for intermolecular cycloaddition via EnT.** A mixture of quinoline (51.6 mg, 0.4 mmol, 1 equiv.) and [Ir-1]PF$_6$ (13.5 mg, 3 mol%) in HFIP (2 mL, 0.2 M) was degassed by sparging with argon for 5 min in a 10 mL Schlenk tube. After adding 1-hexene (101 μL, 0.8 mmol, 2 equiv.), the reaction was stirred and irradiated with 18 W blue LEDs. After the reaction, 18 mL HFIP was added to the solution to obtain a diluted mixture. The mixture was then filtered through the COF Tp-TAPB membrane under an argon atmosphere (2 bar). After the nanofiltration, the COF membrane, together with the retentate, was washed with EtOAc (5×10 mL) under sonication for 30 min. The resulting solution was analyzed by UV-Vis absorption to calculate the recovery rate of the [Ir-1]PF$_6$. The solution was concentrated by rotary evaporation and then dried under vacuum. The recovered [Ir-1]PF$_6$ was used for the next cycle. The permeate solution after nanofiltration was evaporated under reduced pressure. Dibromomethane (14 μL, 0.2 mmol) was added to the residue for crude $^1$H NMR measurement in CDCl$_3$ to determine the yield. Ten cycles of reactions and separations were conducted. After the final cycle, the recovered catalyst was dried and analyzed by NMR measurements.

**General procedure for intramolecular cycloaddition via EnT.** A mixture of 1-(2-vinylphenyl)-propenone (158 mg, 1 mmol, 1 equiv.), [Ir-1]PF$_6$ (22.4 mg, 2 mol%), and DCM (50 mL, 0.02 M) in a 100 mL RBF was degassed by sparging with argon for 5 min in an ice/water bath. The solution was stirred vigorously under irradiation with 24 W white LEDs. After 24 h of reaction, the mixture was filtered through the COF Tp-TAPB membrane under an argon atmosphere (2 bar). After the nanofiltration, the COF membrane, together with the retentate, was washed with EtOAc (5×10 mL) under sonication for 30 min. The resulting solution was analyzed by UV-Vis absorption to calculate the recovery rate of the photocatalyst. Then the solution was concentrated by rotary evaporation and dried under a vacuum. The recovered [Ir-1]PF$_6$ was used for the next cycle. The permeate solution after nanofiltration was evaporated under reduced pressure. Dibromomethane (70 μL, 1 mmol) was added to the residue for crude $^1$H NMR measurement in CDCl$_3$ to determine the yield. A total of 10 cycles of reactions and separations were conducted. After the final cycle, the recovered catalyst was dried and analyzed by NMR measurements.

**General procedure for alkylation of heteroarenes via SET.** Fasudil monohydrochloride (327 mg, 1 mmol, 1.0 equiv.), [Ir-2]PF$_6$ (13.7 mg, 1.5 mol%), p-toluenesulfonic acid monohydrate (TsOH·H$_2$O, 570 mg, 3 mmol, 3.0 equiv.), 16 mL MeOH, and 4 mL DMSO were added into a 50 mL Schlenk tube. The reaction mixture was degassed by sparging with argon for 10 min. After adding ethyl 2-mercaptopropionate (5 mol%), the mixture was irradiated with 2×40 W blue LED at room temperature with a fan for cooling. After 48 h of reaction, the mixture was filtered through the COF Tp-TAPB membrane under an argon atmosphere (2 bar). After the nanofiltration, the COF membrane,

together with the retentate, was washed with EtOAc (5×10 mL) under sonication for 30 min. The resulting solution was analyzed by UV-Vis absorption to calculate the recovery rate of photocatalyst. Then the solution was concentrated by rotary evaporation and dried under vacuum. The recovered [Ir-2]PF$_6$ was used for the next cycle. The permeate solution was diluted with 1 M sodium hydroxide aqueous solution (4 mL) and DCM (30 mL), washed with brine (3×10 mL), dried over sodium sulfate, and concentrated under reduced pressure. Dibromomethane was added to the residue for crude $^1$H NMR in CDCl$_3$ to determine the yield. Ten cycles of reactions and separations were conducted. After the final cycle, the recovered catalyst was dried and analyzed by NMR measurements.

**General procedure for aromatic C – H thiolation via SET.** Under an argon atmosphere, a mixture of N-phenylbenzothioamide (106.5 mg, 0.5 mmol, 1 equiv.), Co$^{III}$(dmgH)$_2$(4-NMe$_2$Py)Cl (17.8 mg, 8 mol%), [Ru](PF$_6$)$_2$ (15 mg, 3.5 mol%), Na$_2$CO$_3$ (64 mg, 0.6 mmol, 1.2 equiv.), H$_2$O (200 μL), and degassed dry MeCN (5 mL) in a 10 mL Schlenk tube were stirred under irradiation of 18 W blue LED for 24 h. After completion of the reaction, the solid was filtered off by filter paper and washed with 5 mL MeCN. The resulting homogenous solution was filtered through the COF Tp-HZ membrane under an argon atmosphere (4 bar). After nanofiltration, the COF membrane, together with the retentate, was washed with EtOAc (5×10 mL) under sonication for 30 min. The resulting solution was analyzed by UV-Vis absorption to calculate the recovery rate of photocatalyst. Then the solution was concentrated by rotary evaporation and dried under vacuum. The recovered [Ru](PF$_6$)$_2$ was used for the next cycle. The permeate solution after nanofiltration was evaporated under reduced pressure. The yield was analyzed through crude $^1$H NMR measurement in CDCl$_3$ using 1,3,5-trimethoxybenzene (28 mg, 0.167 mmol) as an internal standard. Ten cycles of reactions and separations were conducted. After the final cycle, the recovered catalyst was dried and analyzed by NMR measurements.

**General procedure for enantioselective Minisci-type addition via SET.** A mixture of 4-methylquinoline (338 μL, 2.5 mmol, 1.0 equiv.), 1,3-dioxoisoindolin-2-yl acetyl-L-phenylalaninate (969 mg, 2.75 mmol, 1.1 equiv.), [Ir-1]PF$_6$ (42 mg, 1.5 mol%), (R)-TRIP (141 mg, 7.5 mol%), and 1,4-dioxane (25 mL) in a 100 mL RBF was degassed by sparging with argon for 5 min. The reaction was irradiated with 18 W 456 nm blue LEDs and maintained at room temperature with a fan. After 14 h, the mixture was filtered through the COF Tp-TAPB membrane under an argon atmosphere (2 bar). After nanofiltration, the COF membrane, together with the retentate, was washed with EtOAc (5×10 mL) under sonication for 30 min. The resulting solution was analyzed by UV-Vis absorption to calculate the recovery rate. Then the retentate was concentrated by rotary evaporation and dried under vacuum. The recovered [Ir-1]PF$_6$ and (R)-TRIP were used for the next cycle. The permeate solution after nanofiltration was evaporated under reduced pressure. Dibromomethane (175 μL, 2.5 mmol) was added to the residue for crude $^1$H NMR measurement in CDCl$_3$ to determine the yield. Ten cycles of reactions and separations were conducted. After the final cycle, the recovered catalysts were dried and analyzed by NMR measurements.

### Large-scale photocatalyst recovery

(S)-Methyl-2-amino-4-methylpentanoate sulfate (22.1 g, 90 mmol), N-fluorobenzenesulfonimide (NFIS, 1.5 equiv.), and sodium decatungstate (NaDT, 2.21 g, 1 mol%) were dissolved in 500 mL mixed solution of MeCN-H$_2$O (9:1) then degassed by argon sparging for 15 min. The homogenous reaction mixture was pumped via syringe pump through a photoreactor (3×40 W 370 nm Kessil lights, total volume of 6 mL, 1/16 inch I.D. tubing) at 120 μL/min (residence time of 50 min) and collected in a receiving bottle. The $^1$H NMR yield was analysed by

taking 0.3 mL sample diluting with 0.3 mL $CD_3CN/D_2O$ (9/1). After the reaction, the collected mixture was filtered through the COF Tp-DHBD membrane under argon (4 bar). After nanofiltration, the COF membrane, together with the retentate, was washed with water (5×20 mL) under sonication for 1 h. The recovered catalyst was obtained by vacuum drying and weighted to calculate the recovery rate (recovered NaDT 2.1 g, 95% recovery rate). The permeate solution was concentrated to around 100 mL and dried by azeotropic distillation with 2-MeTHF (2×200 mL). The product was precipitated by adding 2-MeTHF (500 mL), filtered, washed with 2-MeTHF, and dried under a nitrogen stream. (S)-methyl 2-amino-4-fluoro-4-methylpentanoate sulfate was isolated as a white amorphous solid with a yield of 90%.

### Stepwise separation of photocatalysts and products

1-Adamantane trifluoroborate potassium salt (72.6 mg, 0.3 mmol, 1.5 equiv.), tert-butyl 4-((4-bromophenyl)sulfonyl)piperazine-1-carboxylate (81 mg, 0.2 mmol, 1 equiv.), $K_2HPO_4$ (69 mg, 0.4 mmol, 2 equiv.), Ni(bpy)$Br_2$ (3.8 mg, 5 mol%), and [Ir-3]$PF_6$ (4.0 mg, 2 mol%) were added into a 10 mL Schlenk tube. The tube was sealed, evacuated, and backfilled with argon three times. After adding 2-((1 S,5 R)-6,6-dimethylbicyclo[3.1.1]hept−2-en-2-yl)ethyl acrylate (66 mg, 0.3 mmol, 1.5 equiv.) and 2 mL degassed THF, the reaction was irradiated with 18 W, 456 nm blue LEDs for 12 h at room temperature. The mixture was passed through a pad of Celite® and eluted with THF to remove the $K_2HPO_4$ salt. The yield was analyzed through crude $^1H$ NMR measurement in $CDCl_3$ using dibromomethane as an internal standard.

After the reaction, the resulting solution was totally diluted to 10 mL and filtered through a COF Tp-TAPB membrane under argon (4 bar) to selectively separate the photocatalyst [Ir-3]$PF_6$ from other components. After nanofiltration, the COF membrane, together with the retentate, was washed with THF (5×10 mL), and the resulting solution was analyzed by UV-Vis absorption to calculate the recovery rate of [Ir-3]$PF_6$. The recovered catalyst was dried and analyzed by NMR measurements. The permeate solution from the first-step nanofiltration was further filtered through a COF Tp-HZ membrane under argon (4 bar) to purify the product from the reactant residues. After nanofiltration, the COF membrane, together with the retentate, was washed with THF (5×10 mL) under sonication for 30 min. The yield of the product was analyzed through the crude $^1H$ NMR in $CDCl_3$ using dibromomethane as an internal standard. The purity of the product was analyzed by HPLC.

## Data availability

The source data underlying Figs. 1d, e, 2b, and 3a–f are provided as a Source Data file. The single-crystal XRD data set for NaDT is available in the Cambridge Crystallographic Data Center (CCDC 2189883). The data that support the findings of this study are available from the corresponding authors upon request. Source data are provided with this paper.

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

## Acknowledgements

This research was supported by the Ministry of Education - Singapore (MOET2EP10120-0014 (J.W.), MOE2019-T2-1-093 (D.Z.), MOE-T2EP10122-0002 (D.Z.)), the Energy Market Authority of Singapore (EMA-EP009-SEGC-020, D.Z.), the Agency for Science, Technology and Research (AME IRG Grant A20E5c0096 (J.W.), U2102d2004 (D.Z.), U2102d2012 (D.Z.)), and the National Research Foundation (NRF-CRP25-2020RS-0002 (J.W.), NRF-CRP26-2021RS-0002 (D.Z.)).

## Author contributions

D.Z. and J.W. conceived the research. H.Y. and J.X. designed the experiments. H.Y. fabricated carbonized PAN substrates and COF membranes and accomplished their optimization and characterization. J.X. and H.C. conducted the photocatalysis reactions. H.Y., J.X., and H.C. performed the catalyst recovery and analyzed the data. D.Z and J.W. funded and supervised the whole research. All authors contributed to writing and revising the manuscript.

## Competing interests

The authors declare the following competing financial interest(s): Two patents have been filed by the National University of Singapore based on the present results (SG Non-Provisional Application No. 10202108259V, SG Non-Provisional Application No. 10202108256S). The authors declare no further competing interests.
