## [Peer Review File · Nature Communications]

Recovery of Homogeneous Photocatalysts by Covalent Organic Framework MembranesReviewers' Comments:

Reviewer #1:

Remarks to the Author:

The authors report the extremely good performance of supported COF membranes in the recovery of homogeneous catalysts, here photo-catalysts. The authors present a very complex study with such facets like non-swelling of the support, a clever cascade separation which uses the modular building plan concept of COFs, stability and solvent-resistance of the supported COF membranes. The authors even present the stable performance of the recovered photo-catalysts in different photo-catalytic reactions as a function of the number of recyclings.

This is a very careful and complex experimental work. I cannot see any major weakness. However, in a revision, authors should address the following points.

- Authors claim that this paper is the "first example of photo-catalyst recovery by COF membranes". This is true, but a rather narrow claim. In Abstract and Conclusions, authors should claim the general recovery of homogeneous catalysts. The authors should mention in more detail the advantages of COFs in comparison with other polymeric nanofiltration membranes. Nanofiltration is a promising technology in preparing drinking water from ground water thus separating pharma-residues (diclo, ibuprofen) hormones etc. Are the reported COF membranes also useful in hydrophilic nanofiltration, when water is sent through the COF membrane?
- COFs can separate macromolecules from aqueous and non-aqueous solvents. Which other separation techniques exist to recover the valuable organic homogeneous catalysts? Compare stability, price, regeneration of fouled membranes...
- The role of photochemistry is a little bit overestimated: "Using light as a form of clean and renewable energy to promote organic chemical synthesis has witnessed drastic developments over the past two decades". There are no big processes using solar or artificial light in chemical synthesis. Also the destruction of organic molecules by photo-catalysis using TiO₂ has not been commercialized so far.

Reviewer #2:

Remarks to the Author:

This work describes the use of COF membranes to recover homogeneous photocatalysts from the reaction mixture. The recovery is pretty good and holds promise for the field of photocatalysis. However, I do miss experiments where the catalyst is recycled in a continuous and integrated fashion, like what is typically done for state-of-the-art nanofiltration studies (see review by van Koten cited by the authors). Now the catalyst is trapped in the COF itself and then released by dissolution with a suitable solvent. All these steps are carried out individually which cost significant time and effort, and thus no real benefit compared to the state of the art can be observed. As a consequence, I think this work is more focused on the advancement of COF than on the improvement of photocatalytic transformations or the recovery itself. If the advancement in the field of COF is notable (I cannot judge that as I am not a COF expert), then the paper might be suitable for publication in Nature Commun. However, if that is not the case, then I do not see a substantial enough advancement that would merit publication in Nature Commun. Some more detailed remarks are as follows:

- The authors have missed this recent and relevant work by the Noel group on the recovery and recycling of photocatalysts in their introduction: Nature Communications (2022) 13:6147

- The work of the authors describes a reaction, catalyst filtration, catalyst recovery and then reuse. None of the steps are connected like shown in normal nanofiltration studies. More specifically, the COF is used only for separating the catalyst from the reaction mixture but actually permanent recycling is not achieved like in the recent Nature Commun paper described in the previous point. Consequently, I see this work as a filtration of photocatalysts and not really a recovery study. E.g., the large scale reaction has been done with a large amount of catalyst. However, recovery and immediate recycling

to the photo reactor should allow for a significant reduction of the total catalyst amount.

- I am a bit skeptical with the results obtained with Ru- or Ir-based photocatalysts. It is known that these catalysts degrade under the photocatalytic reaction conditions. See this work from the Stephenson group: Chem. Sci. 2015, 6, 537-541. Stephenson et al. showed that the radicals actually attack the ligands of the photocatalyst which is the cause of the photodegradation. This process is fast and should be noticeable via NMR and even UV-VIS. Since the steps in this work are individually carried out, I assume that the same type of degradation also should occur in the authors their work. Maybe the degraded photocat (i.e., with modified ligands) might be too large and is not stuck in the cavity of the cat. Hence, the losses observed by the authors can be maybe associated to this specific aspect. Consequently, I think the authors should do NMR studies on the recovered photocatalyst (especially for cats that can undergo degradation, like Ru, Ir based ones + organic dyes).

- Based on this, the title of the manuscript should also be adjusted. Instead of recycling, it should say recovery. The membrane actually does not partake in the recycling.

While the work is overall interesting it still misses the connection between the different steps which is required for a high impact journal. I do not see the advancement that the authors claim to have made. Hence, I think this paper is more suited for a more specialized journal such as Chem Commun, Green Chemistry or ChemSusChem.

Reviewer #3:

Remarks to the Author:

This paper presents the preparation of imine-linked COF membranes on carbonized PAN substrates. These membranes were characterised, and their performance found to be superior for organic solvent nanofiltration compared to commercially available and literature reported TFC and graphene oxide membranes. Optimum COF membranes for direct separation of widely used homogeneous photocatalysts were identified, and successfully applied for catalyst recycling in a range of useful photocatalytic transformations. Catalyst recovery and activity were impressively maintained for 10 cycles in all cases. Stepwise separation using COF membranes with different pore sizes was also shown to enable catalyst recovery with subsequent product isolation, although the increase in product purity (77% to 86%) was modest and would require further separation. Overall, the advancements reported in this work will have a significant impact on the field of homogeneous photocatalysis, creating new opportunities for industrially relevant and sustainable synthesis. The authors should also be commended for their detailed and well-presented supporting information. I believe this will appeal to the readership of Nature Communications, and is suitable for publication in this journal after addressing the minor comments below.

- Was fouling of the COF membranes observed under any of the reaction conditions? Was a new membrane used between cycles or between reactions? What is the lifetime of the COF membranes? As membrane fouling is important for industrial application, these points need to be discussed.
- The separation performance of COF membranes for specific photocatalysts were conducted in different solvents (Fig 2b). This is used to identify the optimum COF membrane for each photocatalyst, but could this not change depending on the accompanying solvent? For example, Tp-TAPB was the best COF membrane for [Ir-3]PF₆ in THF, but would this be true for all other solvents or is it application specific? This should be clarified.
- Gram-scale catalyst recovery was demonstrated following a continuous-flow reaction. What is the potential of using these membranes for continuous nanofiltration and recycling?

Reviewer #1 (Remarks to the Author):

The authors report the extremely good performance of supported COF membranes in the recovery of homogeneous catalysts, here photo-catalysts. The authors present a very complex study with such facets like non-swelling of the support, a clever cascade separation which uses the modular building plan concept of COFs, stability and solvent-resistance of the supported COF membranes. The authors even present the stable performance of the recovered photo-catalysts in different photo-catalytic reactions as a function of the number of recyclings.

This is a very careful and complex experimental work. I cannot see any major weakness. However, in a revision, authors should address the following points.

1. Authors claim that this paper is the “first example of photo-catalyst recovery by COF membranes”. This is true, but a rather narrow claim. In Abstract and Conclusions, authors should claim the general recovery of homogeneous catalysts. The authors should mention in more detail the advantages of COFs in comparison with other polymeric nanofiltration membranes.

Response: Thanks for the reviewer’s supportive comments and valuable suggestions. According to the reviewer’s suggestion, more detailed descriptions have been added in abstract and conclusion.

Abstract, Page 1, Line 15-17: The description “Recovery of homogeneous catalysts by conventional polymeric membranes is promising but limited, as the modulation of their pore structure and tolerance of polar organic solvents are challenging.” has been added.

Conclusion, Page 17, Line 8-14: The description “The permeance of these COF membranes is higher than that of conventional polymeric membranes by up to two orders of magnitude, making them suitable for scalable separation. The customizable pore sizes allow the COF membranes to be optimized according to the three-dimensional size of the targeted catalyst or compound. These COF membranes also exhibit longer lifetime than conventional polymeric membranes, which are usually intolerant of organic solvents and tend to swell.” has been added.

2. Nanofiltration is a promising technology in preparing drinking water from ground water thus separating pharma-residues (diclo, ibuprofen) hormones etc. Are the reported COF membranes also useful in hydrophilic nanofiltration, when water is sent through the COF membrane?

Response: Yes, these imine-linked COF membranes with hydrophilic -NH₂ groups can also be employed in aqueous nanofiltration. As shown in **Supplementary Figure 12**, we have already examined the removal of dyes from water using prepared COF membranes. The results indicated the potential of these COF membranes for water purification.

Supplementary Figure 12. (a) Dye rejection performance of the COF membranes. (b) UV-Vis absorption spectra of Evans blue in feed and permeate. (c) Photo of feed (left, 100 ppm of Evans blue) and permeate (right).

3. COFs can separate macromolecules from aqueous and non-aqueous solvents. Which other separation techniques exist to recover the valuable organic homogeneous catalysts? Compare stability, price, regeneration of fouled membranes.

Response: Other separation techniques to recover homogeneous catalysts include immobilization, distillation, and silica gel-based chromatography. The immobilization of photocatalysts on solid resins is normally impractical because of the obstructed light transmission or low catalyst loading capacity of the solid resins. Distillation is energy-intensive and not suitable for the recovery of heat-sensitive homogeneous catalysts. Silica gel-based chromatography may damage the acid-sensitive homogeneous catalysts and consume a large amount of solvent.

Compared with the separation techniques above, membrane separation technology is a more robust, energy-saving, and environmentally-friendly process for the recovery of homogeneous photocatalysts. The relative comparison between membrane separation and other separation techniques has already been discussed in the **Introduction** of the manuscript (**Page 3, Line 1-6**). Besides, the COF membranes in our study showed good anti-fouling and regeneration properties in organic solvents (**Page 13, Line 3-5**).

4. The role of photochemistry is a little bit overestimated: “Using light as a form of clean and renewable energy to promote organic chemical synthesis has witnessed drastic developments over the past two decades”. There are no big processes using solar or artificial light in chemical synthesis. Also the destruction of organic molecules by photo-catalysis using TiO_2 has not been commercialized so far.

Response: Thanks for this comment. We have revised the manuscript for more precise descriptions. The original sentence “Using light as a form of clean and renewable energy to promote organic chemical synthesis has witnessed drastic developments over the past two decades.” has been revised to “Visible-light-induced photocatalysis has witnessed drastic developments in organic chemical synthesis over the past two decades.”

Reviewer #2 (Remarks to the Author):

This work describes the use of COF membranes to recover homogeneous photocatalysts from the reaction mixture. The recovery is pretty good and holds promise for the field of photocatalysis. However, I do miss experiments where the catalyst is recycled in a continuous and integrated fashion, like what is typically done for state-of-the-art nanofiltration studies (see review by van Koten cited by the authors). Now the catalyst is trapped in the COF itself and then released by dissolution with a suitable solvent. All these steps are carried out individually which cost significant time and effort, and thus no real benefit compared to the state of the art can be observed. As a consequence, I think this work is more focused on the advancement of COF than on the improvement of photocatalytic transformations or the recovery itself. If the advancement in the field of COF is notable (I cannot judge that as I am not a COF expert), then the paper might be suitable for publication in Nature Commun. However, if that is not the case, then I do not see a substantial enough advancement that would merit publication in Nature Commun. Some more detailed remarks are as follows:

Response: We sincerely thank the reviewer for the valuable comments on our manuscript. As the reviewer raised concerns about the advancement of this work, we would like to reiterate the novelty and importance.

In this work, we aim to demonstrate advanced COF membrane materials, which possess many advantages compared to conventional polymeric membrane materials, for homogeneous photocatalyst recovery. The commercial membranes cannot tolerate aggressive organic solvents well. Therefore, we provide a general method to prepare robust COF membranes on carbonized polymeric substrates, which promotes the stability of COF membranes to a new level. The resultant COF membranes could tolerate very aggressive organic solvents (such as DMF and DMSO) for more than 30 days, indicating their high potential in industrial applications. Notably, the permeance of these COF membranes is higher than that of conventional polymeric membranes by up to two orders of magnitude.

It needs to be emphasized that this work is the first example of homogeneous catalyst recycling using COF membranes. Moreover, the pore size of the COF membranes can be customized to match the photocatalysts, organocatalysts, and transition-metal catalysts for recovery, suggesting their broad applications in industries.

We did not recycle the catalyst in a continuous and integrated fashion. Instead, we performed dead-end filtration in our study. Although continuous nanofiltration by conventional polymeric membranes has been reported, such as Noël's recent study (Nature Communications (2022) 13:6147), this technology still has many problems with its practical applications, including the match between reaction residence time and membrane separation efficiency, the requirement of high pressure, low flow rate and reaction scale (e.g., 60.5 h continuous operation, < 8 g production, 81% yield in Noël's study; 20 g scale and 90% yield in our study), a narrow scope of tolerated solvent (e.g., in Noël's study, other catalysts such as 4CzIPN, Ru, Ir catalysts were done by dead-end filtration instead of in-line filtration), and membrane fouling after a long period of operation. Moreover, we achieved cascade isolation of an iridium photocatalyst and purification of a small organic molecule product with the COF membrane-based dead-end filtration, which is very difficult to be achieved by continuous in-line filtration. Other benefits of dead-end filtration include solvent-saving and utilization of the recovered catalysts in different reactions.

1. The authors have missed this recent and relevant work by the Noel group on the recovery and recycling of photocatalysts in their introduction: Nature Communications (2022) 13:6147

Response: We apologize for the missing of this reference, as it was published during the submission process of our manuscript. We have included this important study in the revised manuscript as ref. 14.

2. The work of the authors describes a reaction, catalyst filtration, catalyst recovery and then reuse. None of the steps are connected like shown in normal nanofiltration studies. More specifically, the COF is used only for separating the catalyst from the reaction mixture but actually permanent recycling is not achieved like in the recent Nature Commun paper described in the previous point. Consequently, I see this work as a filtration of photocatalysts and not really a recovery study. E.g., the large scale reaction has been done with a large amount of catalyst. However, recovery and immediate recycling to the photo reactor should allow for a significant reduction of the total catalyst amount.

Response: Thanks for this important question. Even though there are a lot of studies for continuous in-line membrane nanofiltration, however, there are still a lot of hurdles for the practical application, and “permanent recycling” is too ideal due to the membrane fouling after a certain range of operation time and tolerance of different organic solvents. For example, in Noël’s study with TBADT catalyzed C-H alkylation, 60.5 h continuous operation was claimed as a long-term reaction with 81% yield of product and less than 8 g production. However, our NaDT-catalyzed C-H fluorination afforded more than 90% product and more than 20 g product isolation, and 95% catalyst was recovered. In Noël’s study with other photocatalysts, such as 4CzIPN, Ru, and Ir catalysts, they conducted dead-end filtration instead of in-line filtration.

We agree with the reviewer that the recovery and immediate recycling of the photoreactor should allow for a significant reduction of the total catalyst amount. However, the continuous in-line separation also limits the reaction scale due to the efficiency of the membrane separation and increases the risk of membrane fouling. In our study, we present a proof-of-concept study to demonstrate for the first time using customizable COF membranes for homogeneous photocatalyst recovery, and the recycling can be done by dead-end filtration, which is easy to operate. The optimization of these COF membranes and practical application in in-line filtration will be the future target in our laboratories.

3. I am a bit skeptical with the results obtained with Ru- or Ir-based photocatalysts. It is known that these catalysts degrade under the photocatalytic reaction conditions. See this work from the Stephenson group: Chem. Sci. 2015, 6, 537-541. Stephenson et al. showed that the radicals actually attack the ligands of the photocatalyst which is the cause of the photodegradation. This process is fast and should be noticeable via NMR and even UV-VIS. Since the steps in this work are individually carried out, I assume that the same type of degradation also should occur in the authors their work. Maybe the degraded photocat (i.e., with modified ligands) might be too large and is not stuck in the cavity of the cat. Hence, the losses observed by the authors can be maybe associated to this specific aspect. Consequently, I think the authors should do NMR studies on the recovered photocatalyst (especially for cats that can undergo degradation, like Ru, Ir based ones + organic dyes).

Response: Thanks for this important question. In general, Ru- or Ir-based photocatalysts are more stable compared to organophotocatalysts. However, the degradation of these metal catalysts may happen depending on the reaction conditions. As in Stephenson’s work mentioned by the reviewer, the electrophilic radical was generated, which was feasible to attack the electron-rich ligands of the photocatalyst, resulting degradation of photocatalysts. However, in our cases, no such radical was generated (via energy transfer or metallaphotoredox), nor rapid intramolecular reaction happened. We did not observe any detectable photodegradation of catalysts in NMR, as the NMR (^1H , ^{31}P , ^{19}F) spectra of the original photocatalysts and recovered photocatalysts after ten cycles are identical (see **Supplementary Figures, 21-23, 32-34, 42-44, 52-54, 62-64, 73-75, 85-87**). In addition, the UV-Vis spectrum of the recovered photocatalyst after ten cycles was the same as the original photocatalyst (**Supplementary Figure 18, 29, 39, 49, 59, 69**). Limited by the length of this response letter, we only

listed the NMR and UV-Vis data of the recovered [Ir-1]PF₆ for the aryl amination reaction below. Other NMR and UV-Vis results mentioned above can be found in the **Supplementary Information**. Also, the high recovery rate and steady performance of the photocatalysts in 10 cycles indicate that the photocatalysts were stable during our experiments. We have already discussed the catalyst degradation issue in the manuscript (**Page 13, Line 7-10**): “The reaction yields in 10 cycles were all steady, indicating the high catalytic activity of the recovered photocatalysts. After the final cycle, the purity of the recovered photocatalysts was confirmed by NMR analysis, showing no degradation of these noble metal catalysts.”

Recovered [Ir-1]PF₆ (after ten cycles)

¹H NMR (400 MHz, Acetone-*d*₆) δ 8.93 (d, *J* = 2.2 Hz, 2H), 8.61 (dd, *J* = 8.5, 2.4 Hz, 2H), 8.40 (dd, *J* = 8.5, 2.4 Hz, 2H), 8.18 (d, *J* = 5.9 Hz, 2H), 7.84 – 7.77 (m, 4H), 6.86 (ddd, *J* = 12.2, 9.3, 2.4 Hz, 2H), 5.97 (dd, *J* = 8.4, 2.4 Hz, 2H), 1.43 (s, 18H).

Original [Ir-1]PF₆

¹H NMR (400 MHz, Acetone-*d*₆) δ 8.94 (d, *J* = 2.0 Hz, 2H), 8.61 (dd, *J* = 8.5, 2.5 Hz, 2H), 8.40 (dd, *J* = 8.8, 2.1 Hz, 2H), 8.18 (d, *J* = 5.9 Hz, 2H), 7.86 – 7.75 (m, 4H), 6.86 (ddd, *J* = 12.3, 9.4, 2.4 Hz, 2H), 5.97 (dd, *J* = 8.4, 2.3 Hz, 2H), 1.43 (s, 18H).

Supplementary Figure 21. ¹H NMR spectra of original and final recovered photocatalyst [Ir-1]PF₆ for the aryl amination reaction.

**Recovered [Ir-1]PF₆
(after ten cycles)**

¹⁹F NMR (377 MHz, Acetone-*d*₆) δ -63.69, -72.67 (d, *J* = 707.2 Hz), -104.75 (d, *J* = 12.0 Hz), -108.07 (d, *J* = 12.1 Hz).

Original [Ir-1]PF₆

¹⁹F NMR (377 MHz, Acetone-*d*₆) δ -63.69, -72.67 (d, *J* = 707.2 Hz), -104.76 (d, *J* = 12.2 Hz), -108.07 (d, *J* = 12.0 Hz).

Supplementary Figure 22. ¹⁹F NMR spectra of original and final recovered photocatalyst [Ir-1]PF₆ for the aryl amination reaction.

Recovered [Ir-1]PF₆
(after ten cycles)

³¹P NMR (162 MHz, Acetone-*d*₆) δ -135.53 -- -152.99 (hept, *J* = 707.13).

Original [Ir-1]PF₆

³¹P NMR (162 MHz, Acetone-*d*₆) δ -135.53 -- -152.99 (hept, *J* = 707.13).

140 120 100 80 60 40 20 0 -20 -40 -60 -80 -100 -120 -140 -160 -180 -200 -220 -240
f1 (ppm)

Supplementary Figure 23. ³¹P NMR spectra of original and final recovered photocatalyst [Ir-1]PF₆ for the aryl amination reaction.

Supplementary Figure 18. UV-Vis spectra of the recovered catalyst for aryl amination reactions in ten cycles.

4. Based on this, the title of the manuscript should also be adjusted. Instead of recycling, it should say recovery. The membrane actually does not partake in the recycling.

Response: Thanks for this insightful suggestion. We have revised the title of the manuscript to "Recovery of Homogeneous Photocatalysts by Covalent Organic Framework Membranes".

While the work is overall interesting it still misses the connection between the different steps which is required for a high impact journal. I do not see the advancement that the authors claim to have made. Hence, I think this paper is more suited for a more specialized journal such as Chem Commun, Green Chemistry or ChemSusChem.

Response: We appreciate that the reviewer considered our work overall interesting. Once again, we believe the advancement of our work in the field of COF membrane is notable, and the results will be of great interest to the broad readership of *Nature Communications*. Our strategy would be of great economic and social significance. The advancements of our work will have a significant impact on the field of homogeneous photocatalysis, creating new opportunities for industrially relevant and sustainable synthesis.

Based on the reviewer's valuable comment, we added an outlook in the revised manuscript as below:

The outlook "COF membranes have a high potential for continuous nanofiltration and catalyst recycling. However, the integration of COF membranes into continuous catalyst recycling is still immature compared with conventional polymeric membranes. More in-depth studies on using COF membranes for continuous catalyst recycling will be conducted in the future." has been added in the **Conclusion (Page 17, Line 17-21)**.

We deeply appreciate your review of our manuscript and hope that your concerns are fully addressed by our response.

Reviewer #3 (Remarks to the Author):

This paper presents the preparation of imine-linked COF membranes on carbonized PAN substrates. These membranes were characterised, and their performance found to be superior for organic solvent nanofiltration compared to commercially available and literature reported TFC and graphene oxide membranes. Optimum COF membranes for direct separation of widely used homogeneous photocatalysts were identified, and successfully applied for catalyst recycling in a range of useful photocatalytic transformations. Catalyst recovery and activity were impressively maintained for 10 cycles in all cases. Stepwise separation using COF membranes with different pore sizes was also shown to enable catalyst recovery with subsequent product isolation, although the increase in product purity (77% to 86%) was modest and would require further separation. Overall, the advancements reported in this work will have a significant impact on the field of homogeneous photocatalysis, creating new opportunities for industrially relevant and sustainable synthesis. The authors should also be commended for their detailed and well-presented supporting information. I believe this will appeal to the readership of Nature Communications, and is suitable for publication in this journal after addressing the minor comments below.

Response: We sincerely thank the reviewer for the supportive comments.

1. Was fouling of the COF membranes observed under any of the reaction conditions? Was a new membrane used between cycles or between reactions? What is the lifetime of the COF membranes? As membrane fouling is important for industrial application, these points need to be discussed.

Response: Thanks for the reviewer's valuable comments.

Yes, due to the accumulation of photocatalysts on the membrane surface as well as concentration polarization, the fouling of the COF membrane occurred, which can be observed from the permeance decline within filtration time (**Supplementary Figure 19**). However, the COF membranes showed good anti-fouling and regeneration properties in organic solvents, as the initial membrane permeance is steady in all separation cycles. In all cases, the same membrane was used between cycles, while a new membrane was used between different reactions.

Supplementary Figure 19. (a) Permeance of catalyst recovery for the aryl amination reaction as a function of time. (b) Permeance of COF membrane with a feed of constant catalyst concentration.

We conducted at least 10 cycles for the recovery of different catalysts, and the membrane performance was stable. The lifetime of the COF membranes can be higher than 1 month.

According to the reviewer's suggestion, the discussion about membrane fouling has been added in the revised manuscript as shown below:

Page 13, Line 3-5: “The steady permeance in 10 cycles indicates the good anti-fouling properties of COF membranes. Besides, the COF membranes can be easily regenerated with lifetimes higher than one month, demonstrating high potential for industrial application.”

2. The separation performance of COF membranes for specific photocatalysts were conducted in different solvents (Fig 2b). This is used to identify the optimum COF membrane for each photocatalyst, but could this not change depending on the accompanying solvent? For example, Tp-TAPB was the best COF membrane for [Ir-3]PF₆ in THF, but would this be true for all other solvents or is it application specific? This should be clarified.

Response: The separation mechanism of COF membranes is molecular sieving. Thus, the membrane selectivity only depends on the pore sizes of the catalysts, not the solvent. However, the membrane permeance depends on both the pore sizes and solvent properties (e.g., viscosity). Therefore, the optimization of the COF membranes still needs to be considered case by case. The best COF membranes were optimized using the solvent in the corresponding reaction system in our study.

3. Gram-scale catalyst recovery was demonstrated following a continuous-flow reaction. What is the potential of using these membranes for continuous nanofiltration and recycling?

Response: Thanks for this important question. These COF membranes have a high potential for continuous nanofiltration and catalyst recycling. The permeance of these COF membranes could be two orders of magnitude higher than that of conventional polymeric membranes. Compared with conventional polymeric membranes, the integration of COF membranes into continuous catalyst recycling is still immature. For example, the match of residence time and separation efficiency needs to be optimized. A more in-depth study and optimization of COF membranes for continuous catalyst recycling is one target for our future study.

Based on the reviewer’s valuable comment, the outlook “COF membranes have a high potential for continuous nanofiltration and catalyst recycling. However, the integration of COF membranes into continuous catalyst recycling is still immature compared with conventional polymeric membranes. More in-depth studies on using COF membranes for continuous catalyst recycling will be conducted in the future.” has been added in the **Conclusion (Page 17, Line 17-21)**.

Reviewers' Comments:

Reviewer #1:

Remarks to the Author:

I checked carefully the revision and I accept the answers and recommend "accept".

Reviewer #2:

None

Reviewer #3:

Remarks to the Author:

Reviewer 2, statement 3: The authors have carried out NMR and UV-Vis studies to test the stability of the recovered catalyst. Although the NMR results indicate the complexes are stable, there is a clear loss of UV-Vis absorbance with increasing cycle number. Therefore, degradation of catalyst cannot be excluded. The manuscript should be further revised to explain how this is occurring (e.g., the Stephenson work mentioned in the reviewers comments), and the consequence of this deactivation on the recovery and reuse process.

The authors have thoroughly addressed all of the reviewers other comments and made significant improvements to the manuscript. I believe this work will be publishable in Nat. Commun. once the above comment has been addressed.

Reviewer #1 (Remarks to the Author):

I checked carefully the revision and I accept the answers and recommend "accept".

Response: We sincerely thank Reviewer #1 for all the supportive comments.

Reviewer #3 (Remarks to the Author):

Reviewer 2, statement 3: The authors have carried out NMR and UV-Vis studies to test the stability of the recovered catalyst. Although the NMR results indicate the complexes are stable, there is a clear loss of UV-Vis absorbance with increasing cycle number. Therefore, degradation of catalyst cannot be excluded. The manuscript should be further revised to explain how this is occurring (e.g., the Stephenson work mentioned in the reviewer's comments), and the consequence of this deactivation on the recovery and reuse process.

The authors have thoroughly addressed all of the reviewers' other comments and made significant improvements to the manuscript. I believe this work will be publishable in Nat. Commun. once the above comment has been addressed.

Response: We sincerely thank Reviewer #3 for the valuable comments on our manuscript. Since the catalyst recovery rate in each cycle is ~95-99%, the concentration of the recovered catalyst solution (the volume was kept constant for the recovered catalyst solution in each cycle) was gradually decreased with increasing cycle number, resulting in the loss of UV-Vis absorbance. As shown in **Supplementary Figure 18**, the wavelengths of the UV-Vis absorbance of the recovered photocatalyst after ten cycles were the same as the original photocatalyst. Together with the NMR results, it can be concluded that there was no obvious degradation of photocatalysts in our work. Also, the high recovery rate and steady performance of the photocatalysts in 10 cycles indicate that the photocatalysts were stable during our experiments.

Supplementary Figure 18. UV-Vis spectra of the recovered catalyst for aryl amination reactions in ten cycles.

Based on the reviewer's valuable comment, we have further discussed the UV-Vis absorbance loss and the catalyst degradation issue in the revised manuscript (**Page 13, Line 11-23**): "Degradation of photocatalysts would result in low recovery rates and poor performance in the consequent catalyst-reusing processes. In this study, the recovery rates and reaction yields in 10 cycles were all steady, indicating a high catalytic activity of the recovered photocatalysts. After the final cycle, the purity of the recovered photocatalysts was confirmed by NMR analysis (Supplementary Figs. 21 to 23, 32 to 34, 42 to 44, 52 to 54, 62 to 64, 73 to 75, and 85 to 87). No detectable photodegradation of catalysts was observed as the NMR (^1H , ^{31}P , ^{19}F) spectra of the original photocatalysts and the recovered photocatalysts after 10 cycles are identical. In addition, the wavelengths of the UV-Vis absorbance of the recovered photocatalysts after 10 cycles were the same as the original ones (Supplementary Figs. 18, 29, 39, 49, 59, and 69). Therefore, there is no obvious degradation of these noble metal catalysts. It is worth noting that the loss of UV-Vis absorbance with increasing cycle number is due to the slightly decreased concentration of the recovered catalyst solution."

\Reviewers' Comments:

Reviewer #3:

Remarks to the Author:

I am satisfied by the authors explanation regarding catalyst degradation, which is now reflected clearly in the revised manuscript where substantial evidence is provided for their claims.

In my opinion, this work is now acceptable for publication in Nat. Commun.

Reviewer #3 (Remarks to the Author):

I am satisfied by the authors explanation regarding catalyst degradation, which is now reflected clearly in the revised manuscript where substantial evidence is provided for their claims.

In my opinion, this work is now acceptable for publication in Nat. Commun.

Response: We sincerely thank Reviewer #3 for all the supportive comments.